# Reproducibility Report for Reproducibility Challenge 2021

## 1    Reproducibility Summary

### 2    Scope of Reproducibility

In this work we try to reproduce the results in support of the newly proposed model `HierE2E` in Rangapuram et al. [2021], which represents a hierarchical model producing coherent probabilistic forecasts. This new deep learning approach is compared against 11 benchmarks on 5 datasets commonly used in the time series forecasting community. The performance of the different models is measured in terms of the Continuous Ranked Probability Score (CRPS).

### 7    Methodology

The authors provide a well-organized repository at gluonts-hierarchical-ICML-2021. For our experiments we extended their code. Firstly, we made minor modifications to the code written for evaluation of the classical machine learning models. Then, we also added the scripts needed for performing the hyperparameter grid search of the deep learning methods.

### 12    Results

The results we obtain are in alignment with 2 of the 4 central claims made in the original work. We arrive at the same conclusion as the authors that multivariate models such as `DeepVAR` proposed in Salinas et al. [2019] and `DeepVAR+` (`DeepVAR` model modified specifically for analysis purposes in Rangapuram et al. [2021]) outperform the state-of-the-art in the field of hierarchical forecasting. Furthermore, we confirm their claim of `HierE2E` consistently improving along the levels of a hierarchy. Nevertheless, our results do not conclusively show that `HierE2E` is the best model among the evaluated ones. This contrast to their findings holds for both the overall and level-wise CRPS scores.

### 19    What was easy

The data and code used in the experiments were adequately organized in the repository supplementing the paper. Hence, we could readily build upon authors' work.

### 22    What was difficult

Due to resource limitation, we were unable to test certain hyperparameter configurations. However, this omission of configurations was done in such manner that it did not hinder the investigation of the claims made in the paper.

We also came across minor difficulties when installing certain libraries for the test environment proposed by the authors, because our operating system distributions most likely did not match theirs, which was not specified in the repository accompanying the paper.

### 28    Communication with original authors

Before performing the tests, we contacted the corresponding author of the paper to obtain the runtimes of the models. This information was required for allocation of our compute resources. We did not get a response.

# 1  Introduction

The model presented in the paper "End-to-End Learning of Coherent Probabilistic Forecasts for Hierarchical Time Series" improves on the state-of-the-art in the field of hierarchical time series forecasting in multiple respects:

- All currently available models first optimize the loss for the base forecasts, and then a second loss for the reconciliation of these initial forecasts. Differently, `HierE2E` represents an end-to-end model trying to minimize a single loss while accomplishing both tasks.

- Secondly, the proposed model accounts for great flexibility in 3 separate aspects. To begin with, although the tests in the original paper are all conducted on hierarchical datasets, the model allows for general constraints, and not solely hierarchical. Next, the mentioned loss function does not always have to be the CRPS score. Instead, we could use any metric that suits our needs. Lastly, in the first part of the model where we produce the base forecasts, any forecasting method can be used. This means we can plug in either a univariate or multivariate (global) forecaster, a characteristic not common to the entirety of the state-of-the-art.

- The third major contribution that is brought by this model is the fact that it generates probabilistic forecasts. They have gained a great deal of popularity recently due to being instrumental in researchers' or practitioners' decision-making process. So, this is another reason why in the practical world `HierE2E` would be preferred to the competing methods.

# 2  Scope of reproducibility

In the original paper, the above stated features of `HierE2E` were put under investigation on 5 datasets frequently appearing in the hierarchical forecasting literature. The performance of the model was compared against 11 benchmarks outlined in Section 3.1.

The results which were obtained through this evaluation process served the authors make four central claims in the paper:

- **Claim 1**: `HierE2E` outperforms its competitors on all datasets, except on `Tourism`, where it is second best.

- **Claim 2**: The proposed model has the lowest CRPS score (lower is better) at 12 out of 25 total levels (sum of the levels of the 5 datasets). For the remaining 13 levels it comes at the second place with the exception of the 5. level of the `Wiki` dataset.

- **Claim 3**: `HierE2E` achieves performance gains consistently going from the bottom to the top hierarchy levels, unlike some of the state-of-the-art. According to the original work, this should hold for all datasets used in the experiments.

- **Claim 4**: Ablation studies on the `DeepVAR` and `DeepVAR+` models have shown that multivariate models as these two, in spite of not assuming a hierarchical structure in the data, produce better results than the state-of-the-art on the datasets tested. On the other hand, the post-processing during prediction time implemented in `DeepVAR+` increases the CRPS score, hence `DeepVAR` remains the better alternative.

# 3  Methodology

As stated in the **Reproducibility Summary**, the paper we decided to reproduce, was supported by a well-documented repository. Therefore, once having comprehensively studied the paper, we analyzed the code in the repository to be certain it aligns with the model and procedure descriptions in the paper. Subsequently, we familiarized ourselves with the datasets used by performing exploratory data analyses.

After these introductory steps, we proceeded to the test phase. To replicate the paper results for the non-neural models only minor code additions were required. However, in order conduct the hyperparameter grid search explained in the supplementary material of the paper, we had to write a couple of test scripts. These files, the notebooks containing the exploratory data analyses mentioned previously, as well as the code written for the summarization of the experiment results are all accessible at the repository Reproducibility-Challenge-2021.

## 3.1 Model descriptions

In the work of Rangapuram et al. [2021], both classical and deep learning methods are evaluated. For the classical approaches, the base forecasts are obtained in two ways, either from an ARIMA or ETS model. The predictions by these models are made using the R package `hts` [Hyndman et al., 2021] with auto-tuning enabled.

Here we provide brief descriptions of the reconciliation strategies used for the forecasts produced by the ARIMA or ETS models:

- `NaiveBU`: the univariate point forecasts for the bottom-level time series are summed according to the hierarchical constraints to get point forecasts for the aggregate series,
- `MinT` [1] [Wickramasuriya et al., 2019]: general reconciliation procedure that revises unbiased independent univariate base forecasts in such a way that variances of forecast errors are minimized,
- `ERM` [Ben Taieb and Koo, 2019]: a deterministic model which relaxes the unbiasedness assumption of the base forecasts in `MinT` and instead optimizes the bias-variance trade-off of the forecast errors,
- `PERMBU` [2] [Taieb et al., 2017]: utilizes information from different levels in the hierarchy through a sparse forecast combination and a probabilistic hierarchical aggregation.

The main deep learning method investigated in the paper is understandably the proposed model `HierE2E`. As outlined in Section 1, it simultaneously learns from all time series in the hierarchy and incorporates the reconciliation step as part of a single trainable model. The resulting forecasts are both probabilistic and coherent (when predictions for individual series are summed, the hierarchical constraints are not violated).

`HierE2E` allows any kind of base forecaster, but the authors opted for `DeepVAR`. `DeepVAR` is a multivariate, nonlinear generalization of classical autoregressive models. It uses a recurrent neural network (RNN) to exploit relationships across the entire history of the multivariate time series and is trained to learn parameters of the forecast distribution. In the ablation studies, besides `DeepVAR` the authors also evaluate `DeepVAR+`. This is again a `DeepVAR` model, but with the addition of applying reconciliation during prediction time (and not while training the model) as a post-processing measure.

## 3.2 Datasets

The empirical evaluation of `HierE2E` was done on publicly available hierarchical datsets. The datasets as well as the preprocessing scripts are available at gluonts-hierarchical-ICML-2021. In Table 1 summary statistics about the data are presented, and here we offer a brief description of each data source:

- `Tourism` [Tourism Australia, Canberra, 2005] represents a 89-series geographical hierarchy with quarterly observations of Australian tourism flows in the period from 1998 to 2006.
- `Tourism-L` [Wickramasuriya et al., 2019] is an extended version of the dataset `Tourism`. It contains 555 series in a grouped structure, each of them having 228 data points. The series can be structured in two different hierarchies, either based on geography or purpose of travel, both sharing a common root. During the evaluation phase, we calculate the performance of the models for both hierarchies.
- `Labour` [Australian Bureau of Statistics, 2020] contains monthly Australian employment data from February 1978 to December 2020. Based on the category labels provided in the dataset, the authors constructed a 57-series hierarchy.
- `Traffic` [Cuturi, 2011] is a dataset in which the occupancy rate of car lanes on Bay Area freeways is documented. The original sub-hourly data is aggregated following the aggregation strategy elaborated in Ben Taieb and Koo [2019]. This results in a hierarchy of 207 series containing daily observations for one year.

---

[1] The MinT method proposes computing the covariance matrix of individual time series. However, in general, it is difficult to compute this matrix. Hence, in their tests, the authors considered the covariance matrix with shrinkage operator (`MinT-shr`) and the diagonal covariance matrix corresponding to ordinary least squares weights (`MinT-ols`). Both of these matrices are described in more detail in the original work introducing the `MinT` method [Wickramasuriya et al., 2019].

[2] The authors only implement the PERMBU method with MinT reconciliation strategy. Consequently, in the continuation we will refer to this method as PERMBU-MinT.

- Wiki [Google, 2017] includes daily views for 145,000 Wikipedia articles starting from July 2015 to December 2016. The authors restructure the dataset to 150 bottom series and 199 in total, as described in Ben Taieb and Koo [2019].

In addition to the above information, Table 1 also shows the prediction length (number of time steps ahead) $\tau$ used for evaluation of the models on a given dataset. These are the prediction lengths from the publications where the first reference of each dataset occurs.

| Dataset | Total | Bottom | Aggregated | Levels | Observations | $\tau$ |
|---------|-------|--------|------------|--------|--------------|--------|
| Tourism | 89 | 56 | 33 | 4 | 36 | 8 |
| Tourism-L | 555 | 76,304 | 175 | 4,5 | 228 | 12 |
| Labour | 57 | 32 | 25 | 4 | 514 | 8 |
| Traffic | 207 | 200 | 7 | 4 | 366 | 1 |
| Wiki | 199 | 150 | 49 | 5 | 366 | 1 |

Table 1: Summary of the datasets used in the paper. Columns *Total, Bottom and Aggregated* refer to the number of time series. *Observations* reflects the number of data points available for each individual time series in a given dataset. For the Tourism-L dataset the first value in each of the columns *Bottom* and *Levels* is for the geographical hierarchy and the second provides information regarding the hierarchy based on purpose of travel.

## 3.3 Hyperparameters

For the classical machine learning models, the optimal hyperparameters were found automatically, as the forecasters were set to auto-tuning. On the other hand, a hyperparameter grid search following the authors' guidelines in the supplementary material to their paper was performed for the neural models.

The authors tune 5 different hyperparameters:

- batch_size: in HierE2E this hyperparameter takes the value 32 for all datasets except Tourism-L where it is set to 4. Nevertheless, in the optimal hyperparameter sets for DeepVAR and DeepVAR+ the authors had batch_size equal to 32 for Tourism-L. As in all three models the base forecaster is the same, it is unclear why two different values are used for the hyperparameter. Unfortunately, this is not documneted in the repository or explained in the supplementary material, so we repeated authors' experiments.

- context_length_factor: DeepVAR unrolls the RNN over a short subseries of the provided time series, the length of which is known as context_length. In general, this is a multiple (context_length_factor) of the prediction length $\tau$. For the datasets with longer prediction horizons (Tourism, Tourism-L, Labour) the hyperparameter range for context_length_factor was {2, 3, 4}, while for the datasets with shorter prediction horizons (Wiki, Traffic) the range was {15, 25, 40, 60}.

- epochs: in the original work, the tested values of this hyperparameter are: 10, 25, 50 and 100. Due to resource limitation, we do not train the models for 100 epochs. Nevertheless, the number of epochs is not set to 100 in any of the optimal configurations of the deep learning methods reported by the authors in the GitHub repository. Hence, our results are a valid measure for evaluation of their claims.

- num_training_samples: number of samples to draw from the predicted distribution to compute the training loss. The authors tested the performance of the models when obtaining 50 and 200 samples. They report both settings converge to an accurate CRPS score, so we proceeded with the first option.

- warmstart_frac: refers to the fraction of epochs in which no sampling from the learned distribution (this is the second step in HierE2E, which is preceded by producing base forecasts with DeepVAR, and followed by projection of samples to enforce coherency) was done and instead likelihood was optimized directly on the distribution parameters themselves (as in DeepVAR). Two values: 0 and 0.1 are considered for this hyperparameter. As no difference in performance, but only in convergence is reported, we run all configurations for HierE2E with warmstart_frac set to 0.1.

These hyperparameters when taking different values can result in a great deal of configuration combinations. To be more precise, for each of the models DeepVAR, DeepVAR+ and HierE2E on the datasets Tourism, Tourism-L and Labour

we evaluated 9 different configurations, and for the datasets `Traffic` and `Wiki` 12 hyperparameter configurations were put to the test. Moreover, the tests for each configuration comprised of 5 runs.

## 3.4 Experimental setup and code

The evaluation of the models was done through backtesting. For the prediction length $\tau$, let $T + \tau$ be the length of the time series available for a dataset. Then, the first $T$ time steps of a time series are used to tune hyperparameters. Once the best hyperparameters have been determined, all models are trained on $T$ time steps and evaluated on the time steps from $T + 1$ to $T + \tau$.

Important detail in relation to the model evaluation is the metric used to measure performance. Although the `GluonTS` library [Alexandrov et al., 2019] supports calculation of numerous metrics (and in my pickle files with results, all of these metrics are estimated for a given model and dataset), the authors focused on a single well-established metric in the field of probabilistic forecasting - the CRPS score introduced first by Laio and Tamea [2007].

Given a univariate predictive CDF $\hat{F}_{t,i}$ for time series $i$, and a ground-truth observation $y_{t,i}$, CRPS can be defined as

$$CRPS(\hat{\mathbf{F}}_t, \mathbf{y}_t) \coloneqq \sum_i \int_0^1 QS_q(\hat{F}_{t,i}^{-1}, y_{t,i})\, dq, \tag{1}$$

where $QS_q$ is the quantile score (or pin-ball loss) for the $q$-th quantile:

$$QS_q = 2(\mathbb{1}\{y_{t,i} \leq \hat{F}_{t,i}^{-1}(q)\})(\hat{F}_{t,i}^{-1}(q) - y). \tag{2}$$

A discrete version of the metric has been implemented in `GluonTS`, with the integral in Eq. 1 being replaced by a weighted sum over a quantile set. In this work, the quantile set comprised of the quantiles ranging from 0.05 to 0.95 with 0.05 increments.

As aforementioned, the complete code we used for the experiments in this reproducibility study is made available at Reproducibility-Challenge-2021. In the repository, there are also thorough guidelines on replicating our evaluation process.

## 3.5 Computational requirements

The experiments were carried out on two separate machines whose specifications are shown in Table 2. The classical machine learning methods were tested on the personal computer on the left-hand side of Table 2, while the deep learning approaches were evaluated using the other more powerful virtual machine. This machine was run as a notebook instance in Amazon SageMaker [Liberty et al., 2020].

| OS | Linux Ubuntu 20.04 (x64) | Amazon Linux |
|---|---|---|
| CPU | Intel i5-8250U (1.60GHz) | 2.4 GHz Intel Xeon E5-2676 v3 |
| RAM | 8 GB | 64 GB |
| Storage | 300 GB | 5 GB Amazon Elastic Block Store (Amazon EBS) |

Table 2: Specifications of the hardware used for the experiments.

Details about the model runtimes are provided in Table 3. Altogether, the experiment time amounted to **70.03** hours of tests on the personal computer and **173.03** hours of tests on the Amazon SageMaker instance. As it can be observed in Table 3, a siginificant time portion (**112.28** hours or **64.89%**) of the tests run on the Amazon virtual machine was allocated to `HierE2E`. However, these longer runtimes for the newly proposed model do not diminish its improvement over the state-of-the-art shown below in Table 4 and Table 5. Our datasets have daily, monthly and quarterly resolutions meaning we would have sufficient time to train `HierE2E` appropriately.

# 4 Results

In this section we provide the results of our experiments. These have been summarized in Table 4 and Table 5, the former showing the overall performance of the models on different datasets, and the latter offering insights into how models fare in each separate level of a dataset. In Table 5, the classical methods have been considered as a single group. Reporting of the CRPS scores across levels for each model separately has been done in the supplementary material. In the next 4 subsections we analyze the experiment results in terms of the claims stated in Section 2.

| Methods | Labour [min] | Traffic [min] | Tourism [min] | Tourism-L [min] | Wiki [min] | Total [h] |
|---|---|---|---|---|---|---|
| ARIMA-NaiveBU | $2.61 \pm 0.23$ | $9.43 \pm 0.14$ | $0.35 \pm 0.01$ | $10.57 \pm 0.63$ | $2.26 \pm 0.16$ | 2.10 |
| ETS-NaiveBU | $1.73 \pm 0.38$ | $3.42 \pm 0.17$ | $0.32 \pm 0.01$ | $4.11 \pm 0.05$ | $2.11 \pm 0.14$ | 0.97 |
| ARIMA-MinT-shr | $3.07 \pm 0.35$ | $13.18 \pm 5.18$ | $0.45 \pm 0.01$ | $29.48 \pm 0.21$ | $2.47 \pm 0.02$ | 4.05 |
| ARIMA-MinT-ols | $2.69 \pm 0.35$ | $9.71 \pm 0.05$ | $0.47 \pm 0.01$ | $29.51 \pm 0.23$ | $2.46 \pm 0.02$ | 3.74 |
| ETS-MinT-shr | $2.28 \pm 0.11$ | $3.42 \pm 0.01$ | $0.43 \pm 0.02$ | $8.44 \pm 0.19$ | $2.54 \pm 0.07$ | 1.43 |
| ETS-MinT-ols | $2.55 \pm 0.02$ | $3.38 \pm 0.01$ | $0.42 \pm 0.03$ | $8.21 \pm 0.01$ | $2.46 \pm 0.04$ | 1.42 |
| ARIMA-ERM | $24.80 \pm 0.06$ | $13.32 \pm 0.44$ | $6.97 \pm 0.69$ | $435.63 \pm 39.84$ | $3.54 \pm 0.10$ | 40.36 |
| ETS-ERM | $16.25 \pm 0.15$ | $4.27 \pm 0.22$ | $5.91 \pm 0.12$ | $142.75 \pm 17.88$ | $2.98 \pm 0.13$ | 14.35 |
| PERMBU-MinT | $3.15 \pm 0.09$ | $7.40 \pm 0.40$ | $1.99 \pm 0.12$ | - | $6.80 \pm 0.08$ | 1.61 |
| HierE2E | $18.95 \pm 0.01$ | $55.05 \pm 0.40$ | $5.88 \pm 0.03$ | $38.35 \pm 0.61$ | $21.81 \pm 0.02$ | **112.28** |
| DeepVAR | $3.14 \pm 0.05$ | $15.82 \pm 0.14$ | $2.22 \pm 0.04$ | $4.12 \pm 0.07$ | $1.48 \pm 0.02$ | 30.56 |
| DeepVAR+ | $7.75 \pm 0.08$ | $8.25 \pm 0.08$ | $2.11 \pm 0.04$ | $18.55 \pm 0.11$ | $1.52 \pm 0.02$ | 30.19 |

Table 3: In this table we show the time required for a single run of a model with its optimal hyperparameters for a particular dataset. We report the average time in minutes over 5 runs. The model PERMBU_MinT does not work for the dataset Tourism-L, hence no average runtime is shown. In the last column, we present the time required to conduct all our experiments for one of the 12 models evaluated.

| Methods | Labour | Traffic | Tourism | Tourism-L | Wiki |
|---|---|---|---|---|---|
| ARIMA-NaiveBU | 0.0453 | 0.0753 | 0.1138 | 0.1752 | 0.3776 |
| ETS-NaiveBU | 0.0432 | 0.0665 | 0.1008 | 0.1690 | 0.4673 |
| ARIMA-MinT-shr | 0.0467 | 0.0775 | 0.1171 | *0.1615* | 0.2466 |
| ARIMA-MinT-ols | 0.0463 | 0.1123 | 0.1195 | 0.1731 | 0.2782 |
| ETS-MinT-shr | 0.0455 | 0.0963 | 0.1013 | 0.1627 | 0.3622 |
| ETS-MinT-ols | 0.0459 | 0.1110 | 0.1002 | 0.1668 | 0.2702 |
| ARIMA-ERM | 0.0399 | 0.0466 | 0.5885 | 0.5668 | 0.2195 |
| ETS-ERM | 0.0456 | 0.1027 | 2.3742 | 0.5080 | 0.2217 |
| PERMBU-MinT | $0.0393 \pm 0.0002$ | $0.0679 \pm 0.0047$ | **$0.0763 \pm 0.0003$** | - | $0.2790 \pm 0.0200$ |
| HierE2E | **$0.0335 \pm 0.0064$** | *$0.0359 \pm 0.0114$* | *$0.0916 \pm 0.0082$* | $0.1688 \pm 0.0036$ | **$0.1629 \pm 0.0056$** |
| DeepVAR | *$0.0367 \pm 0.0049$* | **$0.0334 \pm 0.0033$** | $0.0953 \pm 0.0056$ | **$0.1394 \pm 0.0019$** | $0.2081 \pm 0.0059$ |
| DeepVAR+ | $0.0457 \pm 0.0116$ | $0.0366 \pm 0.0079$ | $0.0956 \pm 0.0161$ | $0.1979 \pm 0.0263$ | *$0.2053 \pm 0.0130$* |

Table 4: CRPS scores of the methods (with their optimal hyperparameter configurations for a given dataset) averaged over 5 runs. State-of-the-art methods except for PERMBU-MinT produced the same results over all runs, thus no uncertainty is being reported. As already mentioned, PERMBU-MinT does not work for the dataset Tourism-L.

## 4.1 Analysis of Claim 1

Based on the results shown in Table 4, the first claim by the authors is not completely satisfied. Although HierE2E has the lowest CRPS score on the datasets Labour and Wiki, it comes as second on Traffic and Tourism, while it is fifth on Tourism-L.

In terms of the classical approaches, our results align perfectly to the ones reported in the original work. The main discrepancy between our results and the paper results exists in the performance of DeepVAR. As depicted in Table 4, DeepVAR has the best results for the datasets Traffic and Tourism-L and it is second on Labour. As DeepVAR is also a component of HierE2E, this type of scores suggest that DeepVAR might be the main reason for HierE2E outperforming the current state-of-the-art.

## 4.2 Analysis of Claim 2

Regarding **Claim 2** stated above, we also come to different conclusions than the authors. Our results show HierE2E performs the best at 8 levels, and not 12 as reported in the paper. In addition, it is second at 7 of remaining 13 levels, whereas according to the paper, that should be the case 12 times. The explanation for this could again lie in the statement we made earlier about DeepVAR being instrumental to the satisfactory performance of HierE2E. If we look at the scores of DeepVAR in Table 5, we observe that it outperforms other models in 7 situations, and it takes the second place 8

| Datasets | Levels | Best of classical approaches | HierE2E | DeepVAR | DeepVAR+ |
|---|---|---|---|---|---|
| Labour | 1 | 0.0406 ± 0.0004 (PERMBU-MinT) | **0.0302 ± 0.0093** | *0.0342 ± 0.0050* | 0.0445 ± 0.0160 |
| | 2 | 0.0388 ± 0.0003 (PERMBU-MinT) | **0.0342 ± 0.0071** | *0.0362 ± 0.0059* | 0.0461 ± 0.0130 |
| | 3 | 0.0382 ± 0.0002 (PERMBU-MinT) | **0.0335 ± 0.0066** | *0.0362 ± 0.0056* | 0.0456 ± 0.0125 |
| | 4 | *0.0396 ± 0.0003* (PERMBU-MinT) | **0.0361 ± 0.0058** | 0.0403 ± 0.0067 | 0.0466 ± 0.0106 |
| Tourism | 1 | **0.0464 ± 0.0017** (PERMBU-MinT) | 0.0510 ± 0.0099 | 0.0531 ± 0.0120 | *0.0509 ± 0.0190* |
| | 2 | **0.0592 ± 0.0008** (PERMBU-MinT) | *0.0765 ± 0.0113* | 0.0827 ± 0.0091 | 0.0776 ± 0.0216 |
| | 3 | **0.0899 ± 0.0011** (PERMBU-MinT) | *0.1104 ± 0.0080* | 0.1120 ± 0.0086 | 0.1148 ± 0.0180 |
| | 4 | **0.1097 ± 0.0009** (PERMBU-MinT) | *0.1286 ± 0.0079* | 0.1333 ± 0.0062 | 0.1390 ± 0.0152 |
| Tourism-L | 1 | **0.0443** (ARIMA-MinT-shr) | 0.0959 ± 0.0105 | *0.0634 ± 0.0050* | 0.1234 ± 0.0430 |
| | 2 (geo.) | *0.0826* (ARIMA-MinT-shr) | 0.1161 ± 0.0063 | **0.0814 ± 0.0029** | 0.1417 ± 0.0351 |
| | 3 (geo.) | *0.1439* (ARIMA-MinT-shr) | 0.1503 ± 0.0053 | **0.1216 ± 0.0030** | 0.1775 ± 0.0304 |
| | 4 (geo.) | 0.2042 (ARIMA-MinT-shr) | *0.1901 ± 0.0045* | **0.1629 ± 0.0017** | 0.2180 ± 0.0263 |
| | 2 (trav.) | **0.0834** (ARIMA-MinT-shr) | 0.1209 ± 0.0039 | *0.0891 ± 0.0087* | 0.1464 ± 0.0331 |
| | 3 (trav.) | *0.1485* (ARIMA-MinT-shr) | 0.1619 ± 0.0044 | **0.1302 ± 0.0040** | 0.1895 ± 0.0259 |
| | 4 (trav.) | 0.2440 (ARIMA-MinT-shr) | *0.2242 ± 0.0044* | **0.1979 ± 0.0012** | 0.2556 ± 0.0234 |
| | 5 (trav.) | 0.3413 (ARIMA-MinT-shr) | *0.2913 ± 0.0053* | **0.2684 ± 0.0026** | 0.3314 ± 0.0245 |
| Traffic | 1 | **0.0089** (ARIMA-ERM) | 0.0166 ± 0.0170 | *0.0131 ± 0.0058* | 0.0130 ± 0.0081 |
| | 2 | **0.0113** (ARIMA-ERM) | 0.0178 ± 0.0159 | 0.0174 ± 0.0121 | *0.0158 ± 0.0080* |
| | 3 | 0.0254 (ARIMA-ERM) | **0.0186 ± 0.0154** | *0.0198 ± 0.0086* | 0.0209 ± 0.0124 |
| | 4 | 0.1408 (ARIMA-ERM) | *0.0905 ± 0.0061* | **0.0835 ± 0.0027** | 0.0969 ± 0.0096 |
| Wiki | 1 | 0.1558 (ETS-ERM) | *0.0668 ± 0.0056* | 0.0751 ± 0.0153 | **0.0523 ± 0.0158** |
| | 2 | 0.1614 (ETS-ERM) | *0.1184 ± 0.0062* | 0.1199 ± 0.0143 | **0.1053 ± 0.0090** |
| | 3 | *0.2010* (ETS-ERM) | **0.1536 ± 0.0082** | 0.2238 ± 0.0074 | 0.2076 ± 0.0187 |
| | 4 | *0.2399* (ETS-ERM) | **0.1711 ± 0.0067** | 0.2555 ± 0.0109 | 0.2567 ± 0.0205 |
| | 5 | *0.3506* (ETS-ERM) | **0.3047 ± 0.0076** | 0.3663 ± 0.0047 | 0.4047 ± 0.0223 |

Table 5: CRPS scores computed at each aggregation level, again as the average of 5 runs. Level 1 corresponds to the root of a hierarchy. Here we include the results of HierE2E, its 2 variants, and the best classical machine learning approach. The best classical machine learning approach on a particular dataset is the one having the lowest CRPS score at as many levels as possible. In case of ties, the performance of the approach with the best overall CRPS score (averaged over all hierarchy levels) is reported. The best result for a given level is shown in **boldface** and the second best is *italicized*.

times. With regard to the classical approaches, it is important to note PERMBU-MinT manages to beat the deep learning models at each level of the dataset Tourism.

### 4.3 Analysis of Claim 3

**Claim 3** is certainly confirmed by our experiment results. There is only a single exception for the Labour dataset where the CRPS score of HierE2E for the 3. level is lower than the CRPS score for the 2. level. Here we note the same consistency is present in DeepVAR and DeepVAR+, which might lead us to believe this is an inherent characteristic of DeepVAR, and not solely an improvement brought by the newly proposed method.

The authors' statement in terms of the state-of-the-art is supported by our results as well. These methods do not always manage to improve as approaching the higher levels in a hierarchy. Complete level-wise results for each method can be found in the supplementary material.

### 4.4 Analysis of Claim 4

Our results agree with the last claim in the paper. DeepVAR scores better than the state-of-the-art in the hierarchical forecasting field across all datasets. Additionally, DeepVAR+ is superior to these methods on 3 datasets, and is comparable to them on the dataset Labour. However, it has a worse performance than DeepVAR on all datasets, but Wiki where it is marginally better. Hence, it is safe to say that the inclusion of the reconciliation step only when testing the model is not beneficial.

## 5 Discussion

Overall, the experiment results support the statement that `HierE2E` outperforms the classical approaches, and opposed to some of them, also maintains uniform gains along the levels of the hierarchy. Nonetheless, similar observations can be made regarding `DeepVAR` and `DeepVAR+`, with the former empirically proven to be the better option. So, despite the ablation studies in the original work demonstrating `HierE2E` improves on `DeepVAR`, we take the view that `DeepVAR` is the crucial component in `HierE2E`. This is reflected in the results in Table 4 and Table 5. Moreover, even in the cases of datasets where `HierE2E` had a lower mean CRPS score than `DeepVAR`, the uncertainty we report was high enough to not be able to decisively claim the new approach is the optimal one.

The only difference between our experimentation and the one by the authors lies in not training the neural models for 100 epochs. However, as previously mentioned, in the official repository for the paper, they provide optimal hyperparameter configurations none of which has the `epochs` parameter set to more than 50. The results obtained using these configurations serve them do the analysis in the paper. Consequently, our hyperparameter grid search and the experiments as a whole can be considered valid when assessing the claims in the original paper.

### 5.1 What was easy

The data used in the experiments was adequately organized in the repository provided as an addition to the paper. Therefore, we did not experience any issues while carrying out the exploratory data analyses. The same applies to the structure of the code provided as its extension could be done without major problems.

### 5.2 What was difficult

Due to resource limitation, we were unable to test certain hyperparameter configurations. However, as elaborated in the introduction of Section 5, this omission of configurations was done in such manner that it did not hinder the investigation of the claims made in the paper.

We also came across minor difficulties when installing certain libraries for the test environment proposed by the authors, because our operating system distributions most likely did not match theirs, which was unfortunately not specified in the GitHub repository accompanying the paper.

### 5.3 Communication with original authors

As aforementioned, at the beginning of the reproducibility study, we contacted the corresponding author of the paper to obtain the runtimes of the models. The information was required for allocation of our compute resources. We did not get a response.

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
