# OpenReview forum: "Reproducibility Report for Reproducibility Challenge 2021"
_ML_Reproducibility_Challenge/2021/Fall — Reject_

### Official Review · Reviewer_k9S9 · 2022-03-29
**Well verified claims but lack of ablation studies**

**Rating:** 6
**Confidence:** 3

**Review:**

According to the report, the authors have fulfilled all the minimum requirements such as verifying the claims, doing hyper parameter search, and providing their codes. However, I found that the authors were limited to what has already been done in the original paper and focused on them. I found the paper to lack significant ablation studies as well as any recommendation for the original authors. I also found a few grammatical mistakes as well.

So, I believe that this report is on the par with acceptance threshold, but not an excellent one.

---

### Meta-Review · Area_Chair_wko7 · 2022-04-07

**Recommendation:** Reject
**Confidence:** 5

**Metareview:**

Good reproduction effort, with thorough experiments. However, the report lacks thorough discussion on the results, and since the code was available the authors could have perform ablation studies along with hyperparameter search. Thus, I cannot recommend the paper for acceptance.

---

### Decision · Program_Chairs · 2022-04-09

Reject